# ACOUSTIC NEIGHBOR EMBEDDINGS

## ABSTRACT

This paper proposes a novel acoustic word embedding called Acoustic Neighbor Embeddings where speech or text of arbitrary length are mapped to a vector space of fixed, reduced dimensions by adapting stochastic neighbor embedding (SNE) to sequential inputs. The Euclidean distance between coordinates in the embedding space reflects the phonetic confusability between their corresponding sequences. Two encoder neural networks are trained: an acoustic encoder that accepts speech signals in the form of frame-wise subword posterior probabilities obtained from an acoustic model and a text encoder that accepts text in the form of subword transcriptions. Compared to a triplet loss criterion, the proposed method is shown to have more effective gradients for neural network training. Experimentally, it also gives more accurate results with low-dimensional embeddings when the two encoder networks are used in tandem in a word (name) recognition task, and when the text encoder network is used standalone in an approximate phonetic matching task. In particular, in an isolated name recognition task depending solely on Euclidean nearest-neighbor search between the proposed embedding vectors, the recognition accuracy is identical to that of conventional finite state transducer(FST)-based decoding using test data with up to 1 million names in the vocabulary and 40 dimensions in the embeddings.

## 1 INTRODUCTION

Acoustic word embeddings (Levin et al., 2013; Maas et al., 2012) are vector representations of words that capture information on how the words *sound*, as opposed to word embeddings that capture information on what the words *mean*.

A number of acoustic word embedding methods have been proposed, applied to word discrimination (He et al., 2017; Jung et al., 2019), lattice rescoring in automatic speech recognition (ASR) (Bengio & Heigold, 2014), and query-by-example keyword search (Settle et al., 2017) or detection (Chen et al., 2015). Previous works have also applied multilingual acoustic word embeddings (Kamper et al., 2020; Hu et al., 2020) to zero-resource languages, and acoustic word embeddings or acoustically-grounded word embeddings to improve acoustic-to-word (A2W) speech recognition Settle et al. (2019); Shi et al. (2020).

Recently, triplet loss functions (He et al., 2017; Settle et al., 2019; Jung et al., 2019) have been used to train two neural networks simultaneously: an acoustic encoder $f(\cdot)$ network[1] that accepts speech, and a text encoder $g(\cdot)$ network that accepts text as the input. By training the two to transform their inputs into a common space where matching speech and text get mapped to the same coordinates, $f$ and $g$ can be used in tandem in applications where a speech utterance is compared against a database of text, or vice versa. They can also each be used as a standalone, general-purpose word embedding network that maps similar-sounding speech (in the case of $f$) or text (in the case of $g$) to similar locations in the embedding space.

Note that in stricter choices of terminology, one may use the term *acoustic word embedding* to refer only to the output of $g$, and *speech embedding* (Algayres et al., 2020) to refer to the output of $f$.

An obvious application of such embeddings is highly-scalable isolated word (or name) recognition (e.g. in a music player app where the user can tap the search bar to say a song or album title), where

---

[1]It is interesting to note that the basic notion of "memorizing" audio signals in fixed dimensions can be traced back to as early as Longuet-Higgins (1968)

a given speech input is mapped via $f$ to an embedding vector $\mathbf{f}$, which is then compared against a database of embedding vectors $\{\mathbf{g}_1, \cdots, \mathbf{g}_N\}$, prepared via $g$, that represents a vocabulary of $N$ words, to classify the speech. Using the Euclidean distance, the classification rule is:

$$i = \arg \min_{1 \leq j \leq N} ||\mathbf{f} - \mathbf{g}_j||^2, \tag{1}$$

where $|| \cdot ||$ is the $L_2$ norm. Vector distances have been used in the past for other matching problems (e.g. Schroff et al. (2015)). For speech recognition, a rule like (1) is interesting because it can be easily parallelized for fast evaluation on modern hardware (e.g. Garcia et al. (2008)), and the vocabulary can also be trivially updated since each entry is a vector. If proven to work well in isolated word recognition, the embedding vectors could be used in continuous speech recognition where speech segments hypothesized to contain named entities can be recognized via nearest-neighbor search, particularly when we want an entity vocabulary that is both large and dynamically updatable. However, none of the aforementioned papers have reported results in isolated word recognition.

The main contribution in this paper is a new training method for $f$ and $g$ which adapts stochastic neighbor embedding (SNE) (Hinton & Roweis, 2003) to sequential data. It will be shown by analysis of the gradients of the proposed loss function that it is more effective than the triplet loss for mapping similar-sounding inputs to similar locations in a vector space. It will also be shown that the low-dimensional embeddings produced by the proposed method, called Acoustic Neighbor Embeddings (ANE), work better than embeddings produced by a triplet loss in isolated name recognition and approximate phonetic matching experiments. This paper also claims to be the first work that successfully uses a simple $L_2$ distance between vectors to directly perform isolated word recognition that can match the accuracy of conventional FST-based recognition over large vocabularies.

One design choice in this work for the acoustic encoder $f$ is that instead of directly reading acoustic features, a separate acoustic model is used to preprocess them into (framewise) subword posterior probability estimates. "Posteriorgrams" have been used in past studies (e.g. Hazen et al. (2009)) for speech information retrieval. In the proposed work, they allow the $f$ network to be much smaller, since the task of resolving channel and speaker variability can be delegated to a state-of-the-art ASR acoustic model. One can still use acoustic features with the proposed method, but in many practical scenarios an acoustic model is already available for ASR that computes subword scores for every speech input, so it is feasible to reuse those scores.

As inputs to the text encoder $g$, this study experiments with two types of text: phone sequences and grapheme sequences. In the former case, a grapheme-to-phoneme converter (G2P) is used to convert each text input to one or more phoneme sequences, and each phoneme sequence is treated as a separate "word." This approach reduces ambiguities caused by words that could be pronounced multiple ways, such as "A.R.P.A", which could be pronounced as (in ARPABET phones) [aa r p aa], or [ey aa r p iy ey], or [ey d aa t aa r d aa t p iy d aa t ey] (pronouncing every "." as "dot"). In the latter case using graphemes, more errors can occur in word recognition because a single embedding vector may not capture all the variations in how a word may sound. On the other hand, such a system can be more feasible because it does not require a separate G2P.

Also note that while we use the term "word embedding" following known terminology in the literature, a "word" in this work can actually be a sequence of multiple words, such as "John_W_Smith" or "The_Hilton_Hotel" as in the name recognition experiments in Section 5.

## 2 REVIEW OF STOCHASTIC NEIGHBOR EMBEDDING

In short, stochastic neighbor embedding (SNE) (Hinton & Roweis, 2003) is a method of reducing dimensions in vectors while preserving relative distances, and is a popular method for data visualization. Given a set of $N$ coordinates $\{\mathbf{x}_1, \cdots, \mathbf{x}_N\}$, SNE provides a way to train a function $f(\cdot)$ that maps each coordinate $\mathbf{x}_i$ to another coordinate of lower dimensionality $\mathbf{f}_i$ where the relative distances among the $\mathbf{x}_i$'s are preserved among the corresponding $\mathbf{f}_i$'s.

The distance between two points $\mathbf{x}_i$ and $\mathbf{x}_j$ in the input space is defined as the squared Euclidean distance with some scale factor $\sigma_i$:

$$d_{ij}^2 = \frac{||\mathbf{x}_i - \mathbf{x}_j||^2}{2\sigma_i^2}. \tag{2}$$

This distance is used to define the probability of $\mathbf{x}_i$ choosing $\mathbf{x}_j$ as its neighbor in the input space:

$$p_{ij} = \frac{\exp\left(-d_{ij}^2\right)}{\sum_{k \neq i} \exp\left(-d_{ik}^2\right)}. \tag{3}$$

The corresponding "induced" probability in the embedding space is

$$q_{ij} = \frac{\exp\left(-||\mathbf{f}_i - \mathbf{f}_j||^2\right)}{\sum_{k \neq i} \exp\left(-||\mathbf{f}_i - \mathbf{f}_k||^2\right)}. \tag{4}$$

The loss function for training $f$ is the Kullback-Leibler divergence between the two distributions:

$$\mathcal{L}_f = \sum_{i,j} p_{ij} \log \frac{p_{ij}}{q_{ij}}, \tag{5}$$

which can be differentiated to obtain

$$\frac{\partial \mathcal{L}_f}{\partial \mathbf{f}_i} = 2 \sum_j (\mathbf{f}_i - \mathbf{f}_j)(p_{ij} - q_{ij} + p_{ji} - q_{ji}). \tag{6}$$

Hinton & Roweis (2003)'s cogent interpretation of the above equation as "a sum of forces pulling $\mathbf{f}_i$ toward $\mathbf{f}_j$ or pushing it away depending on whether $j$ is observed to be a neighbor more or less often than desired" is a seed for other arguments that will be made in the present paper.

## 3 PROPOSED EMBEDDING METHOD

### 3.1 METHOD DESCRIPTION

Consider a training set of $N$ speech utterances. The $n$'th utterance is characterized by $(S_n, X_n, Y_n)$ where $S_n$ is the audio signal containing one or more words of speech, $X_n$ is a sequence of subword posterior probability vectors for $S_n$, and $Y_n$ is a sequence of subwords pertaining to the reference transcription of $S_n$.

$X_n = [\mathbf{x}_1, \mathbf{x}_2, \cdots, \mathbf{x}_T]$ is obtained from an acoustic model such as a DNN-HMM (Deep Neural Network-Hidden Markov Model) system, where the $d$'th element of $\mathbf{x}_t$ is an estimate of the posterior probability of subword $w_d$ occurring at frame $t$ given the speech signal $S_n$. There is a subword posterior vector for every speech frame, and $T$ is the number of frames in the utterance. While monophone posteriors are used in this study, other subword posteriors could be used, such as HMM state or grapheme posteriors. Also note that raw acoustic features could be used for $X_n$ (see Appendix A where we used MFCCs), but posteriors were used here for reasons mentioned in Section 1.

The subword reference transcription of each utterance is also obtained, such as the phone sequence [k r ao s ih ng], or the grapheme sequence [c r o s s i n g] for the word "crossing," by force-aligning the utterance with its manual word transcription using an ASR with a pronunciation dictionary. This sequence is represented as a sequence of 1-hot vectors $Y_n = [\mathbf{y}_1, \mathbf{y}_2, \cdots, \mathbf{y}_M]$ where $M$ is the number of subwords in the sequence. Each subword usually occupies at least one frame of speech, so $T \geq M$ for every utterance.

Note that the "subwords" used for the posteriors in $X_n$ need not be the same as the "subwords" used for the transcriptions in $Y_n$ (for ANE-g in Section 5, $X_n$ is a monophone posterior vector sequence, whereas $Y_n$ is a grapheme transcription).

The idea in this paper is to train an acoustic encoder neural network $f(\cdot)$ that will transform each posterior vector sequence to a single fixed-dimension embedding vector $\mathbf{f}_n = f(X_n)$ such that predefined relative distances between data samples in the space of $X_n$ will be preserved in the space of $\mathbf{f}_n$ in a manner similar to SNE.

First, since each $X_i$ is a sequence of vectors, the Euclidean distance in (2) does not make sense in our input space. Instead, we define the distance between $X_i$ and $X_j$ based on whether their transcriptions are an exact match in the space of $Y$:

$$d_{ij} = \begin{cases} 0 & \text{if } Y_i = Y_j \\ \infty & \text{else} \end{cases}. \tag{7}$$

Previous work in "supervised SNE"(Cheng et al., 2015) also used an input distance metric incorporating label information. However, they were applied to fixed-dimension point inputs where a sequence of $n$ vectors would be mapped to another sequence of $n$ vectors, not variable-length vector sequences where a sequence of $n$ vectors is mapped to a single embedding vector as in this paper.

It is worth noting that alternate forms of $d_{ij}$ based on dynamic time warping between $X_i$ and $X_j$ (in a manner similar to Hazen et al. (2009)) or $Y_i$ and $Y_j$ with heuristic insertion, deletion, and substitution costs were also tried, but the binary distance above gave much better accuracy. One may expect that smooth "degrees" of dissimilarity in the input space may work better than a hard binary dissimilarity, but there is an intuitive way to understand how the $f$ model can learn such "degrees" from binary labelings as long as there is a sufficient amount of training data. See Appendix B for further discussion on the edit distance as well as justification for the binary distance in (7).

The distance in (7) results in the following probability for (3):

$$p_{ij} = \begin{cases} 1/c_i & \text{if } Y_i = Y_j \\ 0 & \text{else} \end{cases}, \tag{8}$$

where $c_i$ is the number of utterances (other than the $i$'th) that have the same subword sequence $Y_i$.

We use the same induced probability in (4) for the embedding space. Since this probability is explicitly based on the $L_2$ norm, all comparisons we do in the embedding space (e.g. word recognition using (1) or phonetic distance computations in Table 1) are done using the $L_2$ norm.

The loss function for training the acoustic encoder $f$ is $\mathcal{L}_f$ as described in (5). Once we have fully trained $f$, we simply train the text encoder $g$ such that its output for every subword sequence $Y_n$ will match as closely as possible the output of $f$ for the corresponding subword posterior sequence $X_n$, where we keep $f$ fixed. A simple mean square error loss proves to be sufficient for this purpose:

$$\mathcal{L}_g = \sum_{n=1}^{N} ||g(Y_n) - f(X_n)||^2. \tag{9}$$

## 3.2 TRAINING STRATEGY FOR THE $f$ ACOUSTIC ENCODER

In practice, the training data for the proposed system is much larger (millions of utterances) than the original data (thousands of data points) for which SNE (Hinton & Roweis, 2003) was proposed. Hence, we sample the data into equally-sized "microbatches" for computing the loss in (5). Each "microbatch" consists of $N$ data samples, each sample characterized by a subword posterior sequence and subword transcription $(X, Y)$. A fixed number of microbatches then form a minibatch (as in "minibatch training"), and the minibatch loss is the average loss of the microbatches therein. Because the utterances are extremely diverse, forming each microbatch via purely random sampling often results in all the subword sequences being different from each other and $p_{ij} = 0$ everywhere in (8). In such a case, the loss in (5) would be 0, so all gradients would be 0, and the microbatch would have no effect on training. To avoid wasting training time on defunct microbatches, for every microbatch we designate $(X_0, Y_0)$ the "pivot," and artificially search for at least one sample $(X_n, Y_n)$ in the training data that satisfies $Y_n = Y_0$ and insert it in the microbatch. The other samples (with a different subword transcription from $Y_0$) in the microbatch are chosen purely randomly. For further simplicity, we compute the loss over only a subset of $i, j$ pairings in (5), by fixing $i$ to 0:

$$\mathcal{L}_f = \sum_{j} p_{0j} \log \frac{p_{0j}}{q_{0j}}. \tag{10}$$

## 3.3 PHONETIC CONFUSABILITY REFLECTED IN THE **g** VECTORS

In the proposed method, a clear intuition exists on how the Euclidean distance between two embedding vectors $\mathbf{g}_i$ and $\mathbf{g}_j$ can directly reflect the acoustic confusability of their corresponding (purely text) subword sequences $Y_i$ and $Y_j$. Since the text encoder $g$ directly mirrors via (9) the embeddings generated by the acoustic encoder $f$, the **g** vectors – generated purely from text – will mirror the knowledge learned by $f$ on how the text actually sounds.

Consider the two vowels [ae] (as in "bat") and [eh] as in ("bet"). Since they sound similar, it is likely that their posteriors will behave similarly; if one scores high, the other will also score high. On the

Table 1: Euclidean distance between **g** vectors (from ANE-p-40 in Section 5) for different pairs of words, computed by the text encoder $g$ from pure text. The differences in distances are consistent with the intuitive phonetic confusability between the words.

| Words | Phone sequences fed to $g$ network | Distance between **g**'s |
|---|---|---|
| Jackson Jeckson | [j ae k s ah n] [j eh k s ah n] | 0.0478 |
| Jackson Sackson | [j ae k s ah n] [s ae k s ah n] | 1.331 |
| game of thrones game of drones | [g ey m ax f th r ow n z] [g ey m ax f d r ow n z] | 0.122 |
| game of thrones fame of thrones | [g ey m ax f th r ow n z] [f ey m ax f th r ow n z] | 1.077 |

other hand, the consonants [j] and [s] sound distinctly different, and therefore their scores will tend to be opposite of each other; if one scores high, the other will score low, and vice versa.

Now imagine acoustic instances of "Jackson" ([j, ae, k, s, ah, n]), "Jeckson" ([j, eh, k, s, ah, n]), and "Sackson" ([s, ae, k, s, ah, n]) appearing as training samples for $f$. Both the second and third word are only one phoneme away from "Jackson", but the phone posteriorgram sequence for "Jeckson" will be close to that of "Jackson" due to the scores for [ae] and [eh] trending similarly, so their $f$ embeddings will also be inevitably similar. On the other hand, the posteriorgram sequence for "Sackson" will be different from that for "Jackson" because the scores for [j] and [s] tend to be mutually exclusive, so their **f** embeddings will be different.

Now consider the **g** embeddings obtained from pure text inputs "Jackson", "Jeckson", and "Sackson". As we can see in Table 1, "Jackson" is much closer to "Jeckson" than it is to "Sackson" in the **g** embedding space, which is consistent with what actually sounds more similar. Similarly, "game of thrones" is closer to "game of drones" than "fame of thrones."

### 3.4 THE IMPORTANCE OF NORMALIZATION

It is interesting to see what would happen if no normalization were done in (3) and (4). The scores in (8) would be trivial (1 or 0) instead of scaled ($1/c_i$ or 0) binary values, and the computation would be significantly reduced with no denominator in (4). Although this simplicity may be tempting, the resulting gradient is

$$\frac{\partial \mathcal{L}_f}{\partial \mathbf{f}_i} = 4 \sum_j (\mathbf{f}_i - \mathbf{f}_j) p_{ij}. \tag{11}$$

Comparing this with (6) (or later with (12)), we can see that $q_{ij}$ has disappeared entirely from the gradient, meaning the similarity in the embedding space no longer plays a role in weighting the gradient. Only the similarity in the input space ($p_{ij}$) influences the gradient weight, so the fundamental notion of preserving relative distances is no longer being enforced.

## 4 COMPARISON WITH THE TRIPLET LOSS

Some insight into how the proposed method compares with the basic triplet loss criterion, which has been implemented in various forms (Kamper et al., 2016; Settle & Livescu, 2016; He et al., 2017), can be found by inspecting the gradients of the two methods' loss functions used in backpropagation.

For ANE's $f$ network, we can apply (8) to (6) to obtain the gradient of the loss function. The summation can be split into the sum over the samples with the same subword sequence as $i$, i.e., $J_i^+ = \{j : Y_j = Y_i\}$, and the rest of the samples, i.e., $J_i^- = \{j : Y_j \neq Y_i\}$:

$$\frac{\partial \mathcal{L}_f}{\partial \mathbf{f}_i} = 2 \sum_{j \in J_i^+} (\mathbf{f}_i - \mathbf{f}_j) \left( \frac{1}{c_i} - q_{ij} + \frac{1}{c_j} - q_{ji} \right) - 2 \sum_{j \in J_i^-} (\mathbf{f}_i - \mathbf{f}_j)(q_{ij} + q_{ji}). \tag{12}$$

Now, let us consider the triplet loss, placed in the same context as ANE. For every given triplet of posterior sequences $(X_0, X_m, X_n)$ with a corresponding triplet of subword sequences $(Y_0, Y_m, Y_n)$, where $Y_0 = Y_m$ and $Y_0 \neq Y_n$, we have the following basic hinge loss function (He et al., 2017; Settle et al., 2019; Jung et al., 2019):

$$\mathcal{L}_{\text{single-trip}} = \max\left\{0, \alpha - \text{Sim}(\mathbf{f}_0, \mathbf{g}_m) + \text{Sim}(\mathbf{f}_0, \mathbf{g}_n)\right\}, \tag{13}$$

where $\alpha$ is some constant and $\text{Sim}(\mathbf{f}_i, \mathbf{g}_j)$ is some pre-defined similarity function between $\mathbf{f}_i$ and $\mathbf{g}_i$. The idea is to make the embeddings attract each other if their utterances have identical subword sequences, and repel each other if the subword sequences are different.

Note that the state-of-the-art multi-view triplet loss (He et al., 2017; Jung et al., 2019) is actually composed of *two* loss functions as in (21). We will not explore this case, however, as this is beyond the scope of this paper (see Appendix A for further discussion).

In (13), one can see that the $\max$ operator is essentially a "data selection" mechanism. For "light offender" triplets, whose $\text{Sim}(\mathbf{f}_0, \mathbf{g}_m)$ is high and/or $\text{Sim}(\mathbf{f}_0, \mathbf{g}_n)$ is low, the loss is fixed to 0, meaning they do not give rise to any gradients in the training, and are in effect dropped from the training data. Instead, the "heavy offender" triplets, whose $\text{Sim}(\mathbf{f}_0, \mathbf{g}_m)$ is low and/or $\text{Sim}(\mathbf{f}_0, \mathbf{g}_n)$ is high, are the ones kept for training. The margin $\alpha$ controls the border drawn between "light" and "heavy."

To examine what happens with the heavy offenders, we can ignore the $\max$ operator and set $\alpha = 0$ in (13). Let us consider a "batch" triplet loss that is summed over all the samples in our set of $N$ speech utterances:

$$\mathcal{L}_{\text{trip}} = \sum_{i,j} s_{ij} \text{Sim}(\mathbf{f}_i, \mathbf{g}_j), \tag{14}$$

where we have defined

$$s_{ij} = \begin{cases} -1 & \text{if } Y_i = Y_j \\ +1 & \text{else} \end{cases}. \tag{15}$$

Also, since the training tries to make $\mathbf{f}_j$ as close as possible to $\mathbf{g}_j$ for every $j$, we make the approximation $\mathbf{g}_j \approx \mathbf{f}_j$, resulting in

$$\mathcal{L}_{\text{trip}} = \sum_{i,j} s_{ij} q'_{ij}, \tag{16}$$

where $q'_{ij} \triangleq \text{Sim}(\mathbf{f}_i, \mathbf{f}_j)$.

A variety of possibilities exist for $\text{Sim}(\mathbf{f}_i, \mathbf{g}_j)$ in (13). If we assume a Euclidean-distance-based form similar to (4) that is bounded between 0 to 1,

$$\text{Sim}(\mathbf{f}_i, \mathbf{g}_j) \triangleq \exp\left\{-||\mathbf{f}_i - \mathbf{g}_j||^2\right\}, \tag{17}$$

it can be determined that the gradient of the loss in (16) is

$$\frac{\partial \mathcal{L}_{\text{trip}}}{\partial \mathbf{f}_i} = 4 \sum_{j \in J_i^+} (\mathbf{f}_i - \mathbf{f}_j) q'_{ij} - 4 \sum_{j \in J_i^-} (\mathbf{f}_i - \mathbf{f}_j) q'_{ij}. \tag{18}$$

Comparing (18) with (12), we notice a key difference in the contribution of the samples in $J_i^+$ to the gradient. In the case of the triplet loss, the contribution to the gradient is amplified as the similarity $q'_{ij}$ increases, causing more perturbation to the model parameters during backpropagation. This behavior is counterintuitive because a high $q'_{ij}$ means the embedding for $X_i$ and $X_j$ are already almost the same, which is exactly what we're trying to achieve for $j \in J_i^+$, so there is no need to change them further. In contrast, in the ANE loss in (12) one can see in the summation over $J^+$ that as $q_{ij}$ gets higher, we amplify the gradient less and therefore perturb the model parameters less, which is consistent with intuition.

Alternate similarity functions for (17) can also be tried. If $\text{Sim}(\mathbf{f}_i, \mathbf{g}_j) \triangleq -||\mathbf{f}_i - \mathbf{g}_j||^2$, the gradient in (18) becomes $-4 \sum_j s_{ij} (\mathbf{f}_i - \mathbf{f}_j)$, and if $\text{Sim}(\mathbf{f}_i, \mathbf{g}_j) \triangleq \mathbf{f}_i^T \mathbf{g}_j / (||\mathbf{f}_i|| \cdot ||\mathbf{g}_j||)$ (cosine similarity), the gradient is $-2 \sum_j s_{ij} \left\{ \mathbf{f}_i q'_{ij} / ||\mathbf{f}_i||^2 - \mathbf{f}_j / (||\mathbf{f}_i|| \cdot ||\mathbf{f}_j||) \right\}$, and we can make similar arguments as above that they are less effective than the proposed method.

Note that other examples of decomposing loss functions into attractive and repulsive weights can be found in the literature (e.g. Carreira-Perpiñán (2010)).

In summary, both the triplet-loss-based method and the proposed method ANE seem to share the same basic notion of pulling vectors closer if they represent the same word and pushing them apart if not. However, ANE has the added subtlety of pushing or pulling with more "measured strength" based on how good the embeddings currently are. It is possible that part of this effect could be achieved with the triplet loss as well by adjusting the margin $\alpha$ in (13), to change the range of "heavy offender" triplets we train on. Even then, note that the counterintuitive weighting discussed above still holds for the heavy offenders, so the method is rather crude. Previous studies like (Kamper et al., 2016) searched over a range of values to find the best margin. In this regard, it is possible that ANE is more elegant than the triplet loss in that it can find optimal ways to push and pull with less need for additional manual intervention. Data sampling strategies (Riad et al., 2018) have also been proposed to further refine the accuracy of triplet-based embeddings. Whether such strategies would also benefit ANE is a question we leave to future investigation.

## 5    NAME RECOGNITION EXPERIMENTS

Three $f$-$g$ encoder pairs were trained, then tested using the Euclidean nearest-neighbor matching rule in (1). ANE-p is the proposed system using monophone posteriorgram sequences for every $X_n$ and monophone sequences for every $Y_n$ in Section 3.1, and ANE-g is the same but using graphemes for $Y_n$. Trip-p uses the same inputs and outputs as ANE-p but is trained by a triplet distance per Section 4 using the hinge triplet loss in (13) and the bounded form of the $L_2$ distance in (17). For all systems, both $f$ and $g$ were Bi-LSTM networks, with 51 input dimensions (1 dimension for each monophone, including a "silence" monophone) for $f$ and 50 input dimensions (excluding the "silence") for $g$. All $f$ networks had 2 layers and 100 states in each direction, and all $g$ networks had 1 layer and 200 states in each direction (with the exception of ANE-p-40, which had 300 states), with an additional linear layer applied to the last output of every sequence to produce the embedding vector. Training was done using *Adam* (Kingma & Ba, 2015) optimization for all encoders, with an initial learning rate of 0.001.

For training data, speech utterances of varying lengths were extracted from a large set of proprietary data, and their monophone posterior vector sequences and phonetic transcriptions as described in Section 3.1 were obtained using an ASR. The lengths of the utterances were randomly chosen so that the overall distribution of the number of phones per utterance roughly matched that of a database of music titles and person names. The process resulted in a training data set of 4.2M utterances and a development data set (used to stop training) of 1.4M utterances. The average utterance length in the training data was 885 ms (so the mean of $T$ in Section 3.1 was 88.5 frames), and the average number of phones per utterance (the mean of $M$ in Section 3.1) was 8.56. From the training set, 678K microbatches (160 utterances per microbatch, and 32 microbatches per minibatch) were generated for training the $f$ for ANE-p, and 783K microbatches for training ANE-g. From the development set, 130K microbatches were generated for ANE-p, and 141K microbatches for ANE-g. For the triplet-loss-based encoder Trip-p, 17.9M triplets were generated from the training set, and 5.2M triplets from the development set.

The ASR was a "hybrid" DNN-HMM system (Bourlard & Morgan, 1994) with a state-of-the-art convolutional neural network acoustic model (Abdel-Hamid et al., 2014; Huang et al., 2020) accepting 40 filterbank outputs and emitting posteriors for 8,419 HMM states. When training $f$ networks, the 8,419 state posteriors were mapped down to 51 monophone posteriors via a simple uniform mapping based on state cluster membership. The FST decoder was based on Kaldi-OpenFST (Povey et al., 2011; Allauzen et al., 2007) built from a general language model with a 900K vocabulary. Note that this general continuous speech recognizer was used for data preparation and the phonetic matching experiment in Table 4, but *not* for the FST-based recognition in Tables 2 and 3. For the latter, separate whole-word FSTs were built, but the same acoustic model was used.

For the isolated name recognition experiment, 19,646 audio utterances of spoken names were prepared as evaluation data. Each utterance had a corresponding user-dependent list of possible names (i.e., the speaker's "phonebook") of varying size, with an average 1,055 names per phonebook. The phonebook was used to build a whole word FST recognizer for every utterance, using pronunciations obtained from G2P. The same list of pronunciations was used to generate the g embeddings. The FST recognizer was a subword-to-word transducer designed to directly consume subword posterior sequences instead of audio signals, to ensure that the same inputs were used for both the FST and

Table 2: Results of name recognition based on nearest-neighbor search in (1) using the acoustic encoder $f$ and the text encoder $g$ in tandem, compared with the result from a whole-word FST recognizer. ANE-p used $f$ and $g$ trained by the proposed method using monophone transcriptions for each $Y_n$ (see Section 3.1), while ANE-g used grapheme transcriptions for $Y_n$. Trip-p used a triplet distance with the same data as ANE-p. The numeric suffixes indicate the number of dimensions in the embedding vectors.

| Method | Accuracy (%) |
|---|---|
| FST | 97.5 |
| Trip-p-20 | 79.6 |
| Trip-p-30 | 84.7 |
| ANE-g-20 (proposed method) | 96.0 |
| ANE-g-30 (proposed method) | 96.3 |
| ANE-p-20 (proposed method) | 96.9 |
| ANE-p-30 (proposed method) | 97.3 |
| ANE-p-40 (proposed method) | 97.4 |

Table 3: Name recognition results for increasing phonebook size using $f$ and $g$ in tandem. See caption for Table 2 for description of methods. Because the phonebooks are much larger here than in Table 2, the accuracies are lower overall. When the number of dimensions reaches 40, the Euclidean nearest-neighbor search between Acoustic Neighbor Embedding vectors (ANE-p-40) is as accurate as FST decoding.

| Method | Accuracy (%) for phonebook size | | | |
|---|---|---|---|---|
| | 20K | 100K | 500K | 1M |
| FST | 88.1 | 84.4 | 76.8 | 72.1 |
| ANE-g-20 | 79.7 | 75.7 | 67.5 | 62.0 |
| ANE-g-30 | 81.5 | 78.0 | 70.3 | 64.8 |
| ANE-p-20 | 86.5 | 81.7 | 73.1 | 68.2 |
| ANE-p-30 | 88.1 | 84.1 | 76.4 | 71.9 |
| ANE-p-40 | 88.2 | 84.4 | 77.0 | 72.7 |

the $f$ networks. A single acoustic model was used to generate all training and testing inputs for the $f$ networks, as well as the inputs to the FST wherever needed.

Table 2 shows the accuracy of each method in the name recognition task, for varying dimensions in the embedding vectors. For ANE-p and Trip-p, the result of the rule (1) was a best-matching phone sequence, and the match was deemed correct when it was an exact match with any pronunciation in the phonebook for the reference (manually-transcribed) name. For ANE-g, a crude normalization was first done on all names (removing special characters and converting to lowercase) for both training and testing data. The result of the rule in (1) was a best-matching normalized name, and the match was deemed correct when it was an exact match with the normalized reference name. ANE-p was more accurate than ANE-g because ambiguities in word pronunciation were better resolved.

With the triplet distance, competitive accuracy could not be achieved in word recognition. Note, however, that previous studies using the triplet distance generally reported the best accuracy using the cosine distance, not the exponential $L_2$ distance that we used (in a somewhat artificial manner) in (17) to parallel ANE's $L_2$ distance. In Appendix A, we explore further comparisons with the triplet loss in experiments that are more grounded with the existing literature.

In a second experiment, accuracies were measured for user-independent phonebooks of increasing size and are shown in Table 3. Because the phonebook sizes were much larger than in Table 2, accuracy differences can be observed more clearly. For simplicity, a unified phonebook was used for all test samples, where the list contained all the reference names and was increasingly padded with other random names to attain the different sizes in Table 3. There was increasing confusion

Table 4: Approximate phonetic matching accuracy for isolated names using $g$. The FST performs poorly because it uses only a general vocabulary where many of the phonebook names are missing. For ANE-p and Trip-p, the $g$ network was used to transform the FST's 1-best phone sequence into an embedding vector, then a nearest neighbor search was done on the actual phonebook names. See caption for Table 2 for further details on the methods shown.

| Method | Accuracy (%) |
|---|---|
| FST using a default vocabulary | 67.8 |
| Trip-p-20 | 88.8 |
| Trip-p-30 | 89.6 |
| ANE-p-20 (proposed method) | 93.5 |
| ANE-p-30 (proposed method) | 93.9 |

between the names as the phonebook became larger, especially because most of the names were short (e.g. "Lian" and "Layan").

As seen in both Tables 2 and 3, the accuracy of ANE-p increases as the number of dimensions increase because less discriminative information is discarded. For every utterance, the average number of dimensions in the original input used by the FST was 604K (8419 $\times$ the length of the utterance, where the mean length is 71.8 frames). This was reduced to 20, 30, or 40 in Table 3, so the dimension reduction was significant. However, the name recognition accuracy of ANE-p is essentially identical to that of the FST when the number of embeddings is 40.

A third experiment uses the phonebooks in the first experiment to perform recognition via approximate phonetic matching using only the $g$ networks. First, a general large-vocabulary (900K) continuous ASR was run. Since many of the spoken names did not exist in the vocabulary, accuracy is low for the FST in Table 4. Next, from the FST's 1-best phone sequence, a **g** vector was computed, followed by a nearest neighbor search over the **g** vectors of the corresponding phonebook. Accuracy was computed based on how often the correct name was chosen, and ANE-p gave the best result.

The phonetic matching task is "easier" than the word recognition task in the sense that we already obtain a fairly accurate phonetic transcription of the speech from the ASR (as opposed to having to recognize the speech from "scratch"), which is probably why the accuracy of the triplet-based embeddings improve in Table 4 over Table 2. On the other hand, there also exists an "accuracy ceiling" because of some out-of-vocabulary words being transcribed far too wrong by the ASR, which is probably why the high accuracy of ANE in Table 2 drops in Table 4.

## 6 CONCLUSION AND DISCUSSION

This paper has proposed a method of adapting stochastic neighbor embedding (SNE) to sequential data so that spoken words of arbitrary length can be represented as fixed-dimension vectors, called Acoustic Neighbor Embeddings, where the acoustic similarity between words is represented by the Euclidean distance between the vectors.

The proposed training criterion seems to share the same idea as the triplet loss criterion of pulling together vectors in the embedding space that represent the same phonetic sequence and pushing apart vectors that represent different phonetic sequences. To see how the two training criteria in essence differ, the gradients of the two loss functions were inspected to show that the proposed method can be more effective than the triplet loss. The efficacy of nearest-neighbor search using dimension-reduced ANE vectors in isolated name recognition and phonetic matching experiments was also demonstrated. In particular, experiments demonstrated that its accuracy in isolated word recognition can be identical to that of conventional FST-based decoding for vocabulary sizes up to 1 million.

It is possible that creating only one embedding for each phone sequence does not sufficiently capture all the acoustic variabilities in its pronunciation. SNE (Hinton & Roweis, 2003) already has extensions that allow multiple embeddings per input in the dimension-reduced space, which we may be able to leverage in future work to make more robust matches.

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

## A    FURTHER COMPARISONS WITH THE TRIPLET LOSS USING MFCC INPUTS

One goal of this paper has been to examine some differences in the essence of ANE's training criterion in (5) and the triplet distance criterion in (13), not to do a rigorous head-to-head comparison between specific known embodiments or implementations.

However, given that there exists a rich body of work on the triplet distance, it is still meaningful to quantitatively compare ANE with known applications of the triplet loss in experimental settings that are more compatible with the existing literature than Section 5.

One point of interest is that previous methods directly used acoustic feature vectors, rather than posteriorgrams, as the inputs to the $f$-encoder. Another characteristic is that the best results were reported using the cosine distance rather than the $L_2$ distance.

Furthermore, previous methods did not separately train $g$ to explicitly match the output of $f$, as ANE does in (9). It would be interesting to incorporate this idea into previous triplet-based methods.

While word recognition is the main application of interest in Section 5, it is also useful to experiment with the acoustic word discrimination (Kamper et al., 2016) and cross-view word discrimination (He et al., 2017) tasks that have been the main focus of previous literature.

Based on these notions, for this appendix we trained an entirely new set of encoder networks as follows, all using MFCCs for the acoustic inputs and monophones for the text inputs:

1. *mfc-ane-f*: An $f$ encoder trained for MFCC inputs using the same method as described in Section 3.1.

2. *mfc-ane-g*: A $g$ encoder trained using the MSE distance in (9) to mirror the outputs of *mfc-ane-f*.

3. *mfc-tripf-f*: An $f$ encoder trained using a *single-view* triplet distance with the cosine distance on acoustic data only, based on (Kamper et al., 2016; Settle & Livescu, 2016):

$$\mathcal{L}_{\text{single-trip}} = \max\left\{0, \alpha - \text{Sim}(\mathbf{f}_0, \mathbf{f}_m) + \text{Sim}(\mathbf{f}_0, \mathbf{f}_n)\right\}, \tag{19}$$

   where $\mathbf{f}_m$ has the same phone sequence as $\mathbf{f}_0$, $\mathbf{f}_n$ has a different phone sequence from $\mathbf{f}_0$, and $\alpha$ is set to 0.15 as in (Kamper et al., 2016).

   Note that this is probably a more direct parallel of *mfc-ane-f* than the *multi-view* approach of (He et al., 2017) or the triplet-based embeddings used in Section 5, since *mfc-ane-f* is also trained independently of *mfc-ane-g*.

   Since the cosine similarity is stated to give the best result in (Kamper et al., 2016), we too use the cosine similarity in (19).

4. *mfc-tripf-cos-g*: Similar to *mfc-ane-g*, this is a $g$ network that is trained to directly match *mfc-tripf-f*, where the $f$ network is kept constant and only the $g$ network is trained. Note that previous studies have not explored this possibility, but it seems natural to do so in the current context.

   Since *mfc-tripf-f* is based on the cosine distance, *mfc-tripf-cos-g* is also trained to minimize the mean of the cosine distances over a minibatch of "same-word" (in our case, "same-pronunciation") pairs as follows:

$$\mathcal{L}_g = \frac{1}{2N} \sum_{n=1}^{N} \left(1 - \frac{\mathbf{g}_n^T \mathbf{f}_n}{||\mathbf{g}_n|| \cdot ||\mathbf{f}_n||}\right) \tag{20}$$

5. *mfc-tripfg-f*: $f$ network obtained using the multi-view triplet distance of (13) based on (He et al., 2017) using the cosine distance and MFCC inputs.

6. *mfc-tripfg-g*: $g$ network that is jointly trained with *mfc-tripfg-f* via the multi-view triplet distance.

7. *mfc-tripfg-cos-g*: This is a "fine-tuned" version of *mfc-tripfg-g*, where we start with *mfc-tripfg-g* and apply the training criterion in (20) to bring its outputs further closer to those of *mfc-tripfg-f* for "same-word" pairs. Like *mfc-tripf-cos-g*, previous studies have not explored this possibility, but this seems to be a natural extension to try.

Note that the state-of-the-art multi-view triplet-based method is to actually consider *two* (or possibly more) different multi-view triplet losses (He et al., 2017; Jung et al., 2019), of the form

$$
\max\{0, \alpha - \text{Sim}(\mathbf{f}_0, \mathbf{g}_m) + \text{Sim}(\mathbf{f}_0, \mathbf{g}_n)\}
$$
$$
\max\{0, \alpha - \text{Sim}(\mathbf{f}_m, \mathbf{g}_0) + \text{Sim}(\mathbf{f}_n, \mathbf{g}_0)\}
$$

(21)

and to use the sum of the two corresponding batch losses. We do not compare with this loss function in our study, however, because it extends to a realm of multi-view training that is beyond the scope of this paper. It may be possible, for example, to construct a multi-view version of ANE's loss similar to above, which gives ANE the same enhancement as it does to the triplet loss. For example,

$$
\mathcal{L} = \mathcal{L}_1 + \mathcal{L}_2
$$

(22)

where $\mathcal{L}_1$ uses

$$
q_{ij} = \frac{\exp(-||\mathbf{f}_i - \mathbf{g}_j||^2)}{\sum_{k \neq i}(-||\mathbf{f}_i - \mathbf{g}_k||^2)}
$$

(23)

and $\mathcal{L}_2$ uses

$$
q_{ij} = \frac{\exp(-||\mathbf{g}_i - \mathbf{f}_j||^2)}{\sum_{k \neq i}(-||\mathbf{g}_i - \mathbf{f}_k||^2)}.
$$

(24)

This is well beyond the scope of this paper, and we leave as a potential direction in future work.

The data used for this appendix is as follows:

1. Since our purpose is to compare the different embeddings listed above, not to maximize accuracy (i.e., match FST recognition rates in Table 3), we used a significantly reduced data set than that in Section 5 for faster turnaround. The training data and development data consisted of 941K and 335K utterances, respectively. For ANE training, 448K and 186K microbatches (each of size 160) were extracted from the training and development utterances, respectively. For triplet loss training, 4.6M and 1.7M triplet examples were extracted from the training and development utterances, respectively.

2. The inputs to $f$ encoders were MFCCs with 39 features (including delta and delta-delta) per frame, with a frame rate of 100 frames/s and a window length of 25ms. Mean and variance normalization was done based on global statistics from the training data.

3. For testing, from our proprietary data we prepared a set of test data with similar configuration as that used in previous work (Kamper et al., 2016; He et al., 2017). From our evaluation data of random utterances, we extracted 60M pairs of utterances, with mean length 662ms, of which 97K pairs were "same-pronunciation" pairs and the other were "different-pronunciation" pairs. Note that the previous studies actually used "same-*orthography*" or "different-*orthography*" pairs, but this difference would not detract from our goal of comparing different encoders.

The embedding dimensions were fixed to 30 for all $f$ and $g$ networks in this appendix. All $f$ networks were BiLSTM encoders with 2 layers and 400 states per direction per layer, with a 39-dimensional input (for MFCCs) and a final linear layer applied to the final state of the last layer to produce a 30-dimensional embedding. All $g$ networks were BiLSTM encoders with 1 layer and 200 states per direction per layer, accepting 50-dimensional one-hot monophone vector sequences and a final linear layer producing a 30-dimensional embedding. Training was done using *Adam* (Kingma & Ba, 2015) optimization for all encoders, with an initial learning rate of 0.001.

Three experiments were conducted:

1. **Acoustic word discrimination**: Similar to (Kamper et al., 2016; Settle & Livescu, 2016; He et al., 2017), the task was to decide whether two given acoustic sequences represent the same pronunciation or not based on the distance between their $\mathbf{f}$ vectors.

   The $g$ encoders are not used in this task; only the $f$ encoders.

   In the case of *mfc-ane-f*, the $L_2$ distance was used. In the case of *mfc-trip∗-f*, the cosine distance was used. The overall accuracy is measured by the average precision (AP), which is between 0 and 1 and represents the area under the precision-recall curve (the higher the better).

Table 5 shows the results, where we see that *mfc-ane-f* has the highest accuracy among the three embeddings compared.

Among the two triplet-based methods, *mfc-tripf-f* shows a small improvement over *mfc-tripfg-f*. Note that the loss functions used by the two triplet-based methods are essentially the same when we make the approximation that $\mathbf{f}_j \approx \mathbf{g}_j$ as in our analysis in Section 4. The difference is that the loss function for *mfc-tripfg-f* incurs the added "burden" of having to train both the $f$ and $g$ at the same time, and is therefore harder to train than *mfc-tripf-f*. Hence, when all other conditions are held equal (data organization, neural network configuration, optimizer used for training, etc.), it is consistent with our expectations that *mfc-tripf-f* is more accurate than *mfc-tripfg-f*.

2. **Cross-view word discrimination**: Similar to (He et al., 2017), the task is to use $f$ and $g$ in tandem to decide whether a given acoustic sequence and a given text phone sequence represent the same pronunciation or not.

   As with the acoustic word discrimination task, the $L_2$ distance is used with *mfc-ane-\** and the cosine distance is used with *mfc-trip\**, and the overall accuracy is represented by the AP.

   Table 6 shows the results, where *mfc-ane-f* and *mfc-ane-g* provide the highest accuracy.

   Among the two triplet-based methods, *mfc-tripf-f* provides the highest accuracy, consistent with the results in the acoustic word discrimination task.

   With *mfc-tripfg-f*, we can observe a small increase in accuracy by the additional "fine-tuning" step applied in *mfc-tripfg-cos-g*. However, it still scores lower than *mfc-tripf-f* + *mfc-tripf-cos-g*, probably because the latter has a better $f$ encoder.

3. **Name recognition**: This is the same task as Table 2 and the 20K task in Table 3, but using MFCCs instead of posteriorgrams. The Euclidean nearest neighbor match in (1) is used for *mfc-ane-\** embeddings, whereas the cosine nearest neighbor match is used for *mfc-trip\** embeddings:

$$i = \arg \min_{1 \leq j \leq N} \left( 1 - \frac{\mathbf{g}_j^T \mathbf{f}}{||\mathbf{g}_j|| \cdot ||\mathbf{f}||} \right) \tag{25}$$

   Table 7 shows the results, and we can see that the relative accuracy differences are generally similar to those seen in the cross-view word discrimination task in Table 6. These differences get further amplified when we increase the size of the phonebooks to 20K. Again, note that we have used less training data for our encoder networks than those in Tables 2 and 3, since our purpose here is to compare between the different MFCC-based encoders rather than match the accuracy of FSTs.

Table 5: Average precision in acoustic word discrimination task using 30 embedding dimensions

| Method | AP |
|---|---|
| *mfc-ane-f* | 0.622 |
| *mfc-tripf-f* | 0.575 |
| *mfc-tripfg-f* | 0.550 |

Table 6: Average precision in cross-view word discrimination task using 30 embedding dimensions

| $f$ | $g$ | AP (%) |
|---|---|---|
| *mfc-ane-f* | *mfc-ane-g* | 0.829 |
| *mfc-tripf-f* | *mfc-tripf-cos-g* | 0.798 |
| *mfc-tripfg-f* | *mfc-tripfg-g* | 0.772 |
| *mfc-tripfg-f* | *mfc-tripfg-cos-g* | 0.790 |

Table 7: Name recognition accuracy (%) using 30 embedding dimensions on user-dependent phone-books (mean size 1,055) and a static 20K phonebook

| $f$ | $g$ | Variable | 20K |
|---|---|---|---|
| *mfc-ane-f* | *mfc-ane-g* | 88.78 | 64.95 |
| *mfc-tripf-f* | *mfc-tripf-cos-g* | 87.28 | 61.11 |
| *mfc-tripfg-f* | *mfc-tripfg-g* | 82.06 | 51.03 |
| *mfc-tripfg-f* | *mfc-tripfg-cos-g* | 85.30 | 56.64 |

## B  FURTHER DISCUSSION ON THE HARD BINARY DISTANCE IN EQUATION (7)

There is a rich history of computing distances between phonetic sequences or phonetic posteri-orgrams using dynamic programming (see, for example, (Hazen et al., 2009; Ma & Jeon, 2009; Audhkhasi & Verma, 2007)). However, the edit distance tends to invite ad-hoc design decisions that make the process essentially heuristic and brittle. In particular, the substitution, deletion, and insertion costs are difficult to assign in a context-dependent manner. Data-driven phone confusion matrices, for example, can be used to compute substitution costs as in (Audhkhasi & Verma, 2007), but confusion matrices only provide a *context-independent* static cost for any given phone pair that does not account for coarticulation. Additional heuristics are also needed if, for example, one wishes to have a nonlinear length mismatch penalty, as it is hard to achieve this with insertion and deletion costs alone.

As such, we found that using a simple binary distance in (7) results in far more superior models (in terms of recognition accuracy) than an edit-distance-based distance measure.

It can be puzzling that a hard binary distance can result in an $f$ encoder that learns to produce smooth phonetic distances. Note that the triplet distance also shares the same characteristic; in (13), the signs are binary ($+$ or $-$) based on "same" or "different" labelings, with no "degree" of similarity imposed.

Here, we provide some intuition on how the binary distance works.

Consider a pair of audio $(X_i, X_j)$ where $Y_i \neq Y_j$ and therefore we assign $d_{ij} = \infty$. If the words actually have a similar pronunciation, and we have large training data, there will be many other identical pairs in the training data (pairs that are from different audio recordings, but with content almost the same as $(X_i, X_j)$) which have the "same" labeling and are assigned $d_{ij} = 0$. This would have the same effect as $(X_i, X_j)$ being presented to the model *multiple* times, sometimes with the "same" labeling and sometimes with the "different" labeling. The "same"-labeled pairs will counteract the "different"-labels pairs and prevent $q_{ij}$ from becoming 0.

For example, consider the words "cryin'," "crying," and "shouting."

Since "cryin'" and "crying" are very similar, as a toy example we could imagine $2n$ near-identical audio samples where half are labeled "cryin'" and the other half are labeled "crying." The total possible pairs that will get $d = \infty$ from our harsh binary labeling is $n^2$, whereas the total possible pairs that will get $d = 0$ is $2n(n-1)$. Asymptotically, the latter is twice the former, so for any given pair from the $2n$ samples, the expected $q_{ij}$ would be somewhere closer to 1 than 0.

Now consider "cryin'" and "shouting." Because they are so different, for any given $(X_i, X_j)$ it would be extremely hard to find another pair in the training data that is acoustically identical to $(X_i, X_j)$ that is also labeled "same". The distance $d = \infty$ dominates for $(X_i, X_j)$ and the expected $q_{ij}$ would be somewhere close to 0.

Hence, we can intuit that as long as a large amount of training data is used where the *proportion* of "same"-"different" labelings for a given pair of audio is likely to represent the pair's phonetic confusability, ANE will learn smoothed distances that reflects these proportions.

More rigorous mathematical analysis could probably be done in this direction, but in this work we suffice with the above intuitions and leave further quantitative analysis to future work.

