# OpenReview forum: "Acoustic Neighbor Embeddings"
_ICLR.cc/2021/Conference — Reject_

### Official Review · AnonReviewer2 · 2020-10-28
**Novel approach to construct acoustic embeddings with SNE idea behind.**

**Rating:** 6
**Confidence:** 4

**Review:**

While word embedding are very popular and useful for NLP tasks, in speech recognition acoustic embedding plays the same important role (for example to perform query search with the audio recordings). In this paper authors propose a novel idea of acoustic embedding learning, called Acoustic Neighbor Embedding. Using stochastic neighbor embedding (SNE) idea of preserving the relative distances and optimizing Kullback-Leibler divergence (KL), proposed in the paper embedding is constructed optimizing KL divergence (for acoustic) and $L_2$ distance between acoustic embedding and correspondent text embedding (for text). With experiments authors show that their approach is better than triplet loss approach used in the previous works. Also with $L_2$ distance between embedding vectors this paper is the first one which has a competitive isolated word recognition quality compared to the FST-based approaches. Experiments are done for several different scenarios: search for personal phonebook, search in global phonebook of different size and search by closest text embedding after ASR model.

Pros of the paper:
- A new idea for acoustic embedding learning with SNE idea behind.
- Analysis of the proposed loss with the triplet loss.
- Analysis of the phonetic confusability demonstrating how the method behaves for the most complicated cases of recognition.

Cons:
- General application for entity recognition with the proposed approach will be not super easy: besides ASR system there should be also force-aligning system which should extract segments where entities are presented. And the most complicated task is obtaining these force-aligned segments.
- Experiments are done on the private datasets, so results in the paper are not reproducible and not comparable with the previous and future works.
- There is no analysis what is the force-aligning error presented in the data.
- There is no comparison with the other acoustic features instead of "postreriograms" ("posteriograms" could have errors of ASR model itself and possibly not be ideal for acoustic embedding construction) both for triplet loss and proposed method. Previous work He et al., 2017 used MFCC for the triplet loss. Thus to demonstrate that proposed approach is well suited for the task it should be also shown that it outperforms the triplet loss on MFCC input.

Comments:
- "However, none of the aforementioned papers have reported results in isolated word recognition.":  Jung et al., 2019 reports average precision of isolated word recognition (if audio and text are given for the same word). Otherwise, "isolated word recognition" should be clarified in the paper.
- Would like to have an ablation when instead of "posteriograms" raw wave or MFSC/MFCC features are used to investigate 1) what is the effect of ASR system errors on the embedding learning 2) what performance we loose if switch to the "posteriograms" with small network $f(x)$. Experiments for triplet loss should be provided too for this case (it could be that triplet loss behaves better with not "postriograms" input).
- What are the mean and std of the values for $T$ and $M$ in the data (for both monophone and grapheme cases)?
- It is better to move section 3.3 into experiments section 5 otherwise it is not clear with which model $L_2$ distances are computed for examples (Table 1). This is more analysis of a model trained on particular dataset.
- In formulas (13) and (14) the same notation for loss is used $\mathcal{L}_\text{trip}$: the first one is for one sample and the second - for all. Usage of different notation will improve readability.
- Why is silence excluded from the $g(y)$ training? It gives the word boundaries notion so the confusion for the text could occur in the silence absence.
- Could authors clarify what they mean with the "regression layer" (just linear layer?)?
- Comparison with FST baseline is not clear. In case of FST we can generate any sequence of words (say, we have "Katarina F." and "John S." in the phonebook, then FST potentially can infer also "John F." or "Katarina S.", while ANE cannot generate this as we have text embeddings only for original "Katarina F." and "John S.", thus accuracy for FST will be underestimated in this comparison). Could authors clarify this?
- It would be interesting to see accuracy on in-vocabulary and out-of-vocabulary in Table 4 (with the number of such examples) to have the better comparison with FST (showing that in-vocab has the same performance as FST and what is peformance for pure out-of-vocab).

The idea of the paper is very interesting and based on well known and effective SNE approach. The paper itself very well written, with enough clarification to understand the main idea. However, the main concerns of the paper are:
- usage of private data
- lack of the experiments with MFCC input for both triplet loss and proposed loss to show the new method effectiveness (one of the paper claims is that the method is better than triplet loss)
- clarification on the FST evaluation is needed (one of the paper claims that the methods performs similar to FST).

### Edit based on the authors' response

I believe the authors have addressed part of the major concerns that I and the other reviewers had. Comparison with FST approach and triplet loss is clarified and supported by the extensive experiments now, however, most important things are in Appendix only. Based on the updated paper's version I change my rating from "5: Marginally below acceptance threshold" to "6: Marginally above acceptance threshold".

One of the reviewers mentioned about comparison with language model (LM): here we could use character/phoneme based LM and vocabulary which can help to solve ambiguity too. So this could be considered as a good experiment for future work to show the great potential of ANE if it outperforms LM usage.

Still I have several concerns, probably more philosophical from some point of view:
- Rely on the force-aligning data for practical applications (in experiments it was ideal segmentation)
- Usage of private data in experiments.

Authors state "The goal of our paper is to introduce ANE and highlight some interesting things about it.". I don't see any points in the paper and author's comments why this is helpful/applicable/better than some another ideas. Authors mentioned in the comments that they cannot state that ANE is faster than the FST while having the same performance (this could be one good point that we have speed up using small embeddings + simple L2 distance computation as pointed by one the reviewers). About applications for continuous speech authors gave the comment "However, we are not even sure if speech recognition per se is the best application for ANE. We are hoping that the community will find other interesting uses, either with the embeddings themselves or the distances between embeddings.". Thus, I am feeling that the paper is not finished in that respect.

---

> ### Author Response · Authors · 2020-11-19
> **[Part 1/2] Added experiments using MFCCs in Appendix A**
>
> [Part 1/2]
>
> Dear AnonReviewer2, thank you for the helpful review.
>
> Using MFCC inputs is something that we have been interested in as well. The other reviewers have also suggested different variations of the triplet loss (see AnonReviewer6's comments), as well as the fact that previous work reported that the cosine distance, not the L2 distance, gives the best accuracy.
>
> To address these concerns, we have added Appendix A where we trained an entirely new set of encoder networks that take MFCC inputs, in various forms to address some of the issues raised by AnonReviewer6.
>
> We still used private data for training and testing, but conducted word discrimination tasks on a data size that is similar to that of previous work [Kemper 2016, Settle 2016, He 2017], and we believe our implementations are reasonable and our comparative experiments are sound.
>
> Note that our goal is to highlight and contrast the difference in the essence of the two loss functions (ANE vs. triplet), not compare or "compete" between specific implementations of them. We believe our experiments, together with our analysis in Section 4, draw a sufficient picture in this regard.
>
> \>\> There is no analysis what is the force-aligning error presented in the data.
>
> Could the reviewer clarify what they exactly mean here? The encoders trained in this paper all deal with pre-segmented audio, where each entire audio file contains just one word (like "The_Hilton_Hotel"), and it is assumed that the pre-segmentation is correct. If the pre-segmentation is wrong (e.g. the audio contains unwanted speech at the beginning, or is clipped at the end), then this could cause errors in the embeddings, of course, but this is rare, and since all the embeddings are using the same input data, such rare occurrences do not invalidate our comparisons.
>
> \>\> Jung et al., 2019 reports average precision of isolated word recognition (if audio and text are given for the same word). Otherwise, "isolated word recognition" should be clarified in the paper.
>
> [Jung 2019]'s experiment is "cross-view word discrimination". The task is "Given speech X and text Y, determine between hypothesis H1: they represent the same word, or H0: they represent different words". The task is a detection problem.
>
> The task of "Word recognition" is, "given speech X that can be classified into 1 of N classes (w_1, ...,w_N), what is the correct class"? This is a classification problem.
>
> At any rate, we have conducted both the cross-view word discrimination task and the word recognition task on our new set of embeddings in Appendix A.

---

> ### Author Response · Authors · 2020-11-19
> **[Part 2/2; Continued from Part 1]**
>
> [Part 2/2; Continued from Part 1]
>
> \>\> What are the mean and std of the values for T and M in the data (for both monophone and grapheme cases)?
>
> We have added to the paper:
> "The average utterance length in the training data was 885 ms (so the mean of T in Section 3.1 was 88.5 frames), and the average number of phones per utterance (the mean of M in Section 3.1) was 8.56."
>
> \>\> It is better to move section 3.3 into experiments section 5...
>
> To clarify, we added that the g vectors in Table 1 are from ANE-p-40.
>
> \>\> In formulas (13) and (14) the same notation for loss is used ...
>
> Good suggestion. Corrected.
>
> \>\> Why is silence excluded from the g training?...
>
> This is an interesting idea that we did not really think about. This could be tried in future work to see if it helps accuracy.
>
> \>\> Could authors clarify what they mean with the "regression layer" (just linear layer?)?
>
> Yes. We have changed it to "linear layer"
>
> \>\> say, we have "Katarina F." and "John S." in the phonebook, then FST potentially can infer also "John F." or "Katarina S."...
>
> Actually, the FST we used in the name recognition experiments cannot recognize "John F". Every entry in the phonebook is treated as one "word", like "Katarina_F.", "John_S.", "The_Hilton_Hotel", and the FST is a simple finite state grammar that can recognize exactly one such "word". In other words, the FST is built to recognize the exact same set of names as the ANE embeddings.
>
> The general LVCSR FST was used only for data preparation and the approximate phonetic match task in Table 4. It is _not_ the same as the whole-word FST recognizer in Table 2 & 3 (although they use the same acoustic model). We see that this may be a source of confusion for readers, so have clarified this in the text as follows:
>
> "Note that this general continuous speech recognizer was used for data preparation and the phonetic matching experiment in Table 4, but not for the FST-based recognition in Tables 2 and 3. For the latter, separate whole-word FSTs were built, but the same acoustic model was used."
>
> \>\> It would be interesting to see accuracy on in-vocabulary and out-of-vocabulary in Table 4 (with the number of such examples) to have the better comparison with FST (showing that in-vocab has the same performance as FST and what is peformance for pure out-of-vocab).
>
> This is a good suggestion, but it seems we will not have time to do this. The bulk of our effort has been in the experimental validation in Appendix A.
>
> In summary, we believe we have addressed most of your concerns, particularly through our experiments in Appendix A, and ask you to reconsider your score. Please let us know if you have further questions or comments. Thank you.
>
> \- The authors.

---

> > ### Comment · AnonReviewer2 · 2020-11-24
> > **Addressed part of the concerns**
> >
> > Dear authors,
> >
> > Thanks for clarification on FST and force-aligning, and extensive experiments with the MFCC features and triplet loss. I have updated my overall review and changed the decision from "5: Marginally below acceptance threshold" to "6: Marginally above acceptance threshold". Other comments please check in the updated overall review above.

---

> ### Author Response · Authors · 2020-11-24
> **Thanks for your review**
>
> Thank you for your review.
>
> In regard to your last comment, "Thus, I am feeling that the paper is not finished in that respect", please allow us to clarify what we meant by "we are hoping that the community will find other interesting uses".
>
> We refer you to the original papers for other word embeddings like word2vec ("Efficient Estimation of Word Representations in Vector Space" Mikolov 2013) and GloVe ("GloVe: Global Vectors for Word Representation" Pennington 2014).
>
> The original word2vec paper ran experiments on semantic-syntactic word relationship and sentence completion, while the GloVe paper ran experiments on word analogy, word similarity, and NER.
>
> Today, these embeddings are used in many more applications than what those papers experimented on, including machine translation, natural language parsing, word confidence modeling, keyword detection, and many others.
>
> We hope this clarifies what we meant.
>
> Best,
>
> The authors.

---

### Official Review · AnonReviewer4 · 2020-10-29
**Adapting the SNE technique to learn acoustic embeddings; could be more thorough in its experiments.**

**Rating:** 6
**Confidence:** 4

**Review:**

In this work, the authors propose a new training method to learn acoustic embeddings by simultaneously training two encoder networks, one for speech (f) and one for text (g), such that the resulting embeddings from the two networks are in a common subspace. At test time, the embedding for a given speech input using the network f could be compared against a text database of embedding vectors derived using the network g and vice-versa. The proposed technique is shown to be more accurate than a standard triplet loss-based retrieval method and it performs at par with an FST-based speech recognition system.

While the proposed embedding method is a direct adaptation of the stochastic neighbor embedding (SNE) technique by Hinton & Roweis ('03), the modifications have been described very clearly in this draft. The authors also do a nice job of describing why the proposed technique might be more effective than a triplet loss-based method by inspecting the gradients of both losses.

As it stands, the experimental section in the draft is a bit thin:

* For a more direct comparison with prior work on learning acoustic embeddings, it would be useful to see how the proposed embeddings fare on an acoustic word discrimination task (as described in He et al. 2017) where the task is to predict whether a pair of acoustic sequences correspond to the same word or not. Cross-view word discrimination (where the pairs of inputs consist of both written words and spoken words) is another task proposed by He et al. 2017 which could also be easily set up for the proposed method.

* Since embeddings from both f and g are simultaneously learned, could the authors also show experiments on the standard task of spoken term detection where one has to search for a given text query within spoken utterances.

* The authors mention that the use of a hard binary distance as defined in Equation (7) was the best choice, rather than using softer definitions of distance. Some supporting experiments to show how different definitions of the distance function affect performance would be useful. (This could go into an appendix.)

* Given that SNE lends itself well to visualizations, it would be nice for the reader to see visualizations showing how acoustic and textual embeddings of similar words cluster together.

* A relatively minor point. In Section 5, the authors mention that the phonetic transcriptions were obtained using an FST-based ASR with a DNN-HMM acoustic model but don't provide any further details. Please elaborate on the ASR system that was used or include an appendix with more implementation details.

Also, given that the proposed technique performs at par with an FST-based recognizer, the motivation for why the proposed technique is needed should be brought out more clearly. (For example, when using phonebooks of larger sizes, the nearest neighbour search using the proposed embedding method might be more computationally efficient than an FST-based recognizer.)

----------------------

Update after author response:

Thanks to the authors for addressing some of my concerns by conducting additional experiments that are listed in Appendix A. I have now increased my rating from 5 to 6.

---

> ### Author Response · Authors · 2020-11-19
> **Added Appendix A**
>
> Dear AnonReviewer4, thank you for your helpful review.
>
> We agree with you and the other reviewers that further experimental validation should be done. AnonReviewer6 also made some suggestions on enhancing the triplet-based methods to make them more compatible with ANE, and other reviewers also mentioned using MFCCs and the cosine distance, as well as conducting the word discrimination tasks you mentioned.
>
> Hence, we focused our additional efforts on addressing all these needs, and have added Appendix A to our paper, where we trained an entirely new set of encoders. Both the "acoustic word discrimination" task and the "cross-view word discrimination task" in [Kamper 2016, Settle 2016, He 2017] were also conducted, using a data set with a size similar to that in the previous work.
>
> \>\> could the authors also show experiments on the standard task of spoken term detection where one has to search for a given text query within spoken utterances.
>
> The general task of "spoken term detection" is actually extremely involved and we believe is well beyond the scope of this paper. In a set of continuous speech utterances, one must be able to search _portions_ of these utterances at varying start positions and lengths to search for spoken examples of a query text. This involves many complicated issues in segmentation and indexing that we believe is more appropriate for a separate paper.
>
> \>\> The authors mention that the use of a hard binary distance as defined in Equation (7) was the best choice, rather than using softer definitions of distance. Some supporting experiments to show how different definitions of the distance function affect performance would be useful. (This could go into an appendix.)
> \>\> Given that SNE lends itself well to visualizations, it would be nice for the reader to see visualizations showing how acoustic and textual embeddings of similar words cluster together.
>
> These are good suggestions, but because we had to focus all our efforts on strengthening the experimental validation with the triplet loss, it seems we will have to forgo them.
>
> \>\> Please elaborate on the ASR system that was used or include an appendix with more implementation details.
>
> We have added further details on the ASR system to the paper:
>
> "The ASR was a “hybrid” DNN-HMM system (Bourlard & Morgan, 1994) with a state-of-the-art convolutional neural network acoustic model (Abdel-Hamid et al., 2014; Huang et al., 2020) accepting 40 filterbank outputs and emitting posteriors for 8,419 HMM states. When training f networks, the 8,419 state posteriors were mapped down to 51 monophone posteriors via a simple uniform mapping based on state cluster membership. The FST decoder was based on Kaldi-OpenFST (Povey et al., 2011; Allauzen et al., 2007) built from a general language model with a 900K vocabulary. Note that this general continuous speech recognizer was used for data preparation and the phonetic matching experiment in Table 4, but not for the FST-based recognition in Tables 2 and 3. For the latter, separate whole-word FSTs were built, but the same acoustic model was used."
>
> \>\> Also, given that the proposed technique performs at par with an FST-based recognizer, the motivation for why the proposed technique is needed should be brought out more clearly.
>
> Although recognition was a task that we thought is a good way to highlight the quality of the embeddings (in terms of how well they capture phonetic content in an L2 space), we are actually not sure whether recognition per se would be the best application for ANE. Yes, being able to do searches with pure matrix operations (that are highly parallelizable in hardware accelerators) might be nice in certain environments, but in general, FSTs are also very good at pruning unwanted paths and searching fast. The goal of our paper is to introduce ANE and highlight some interesting things about it. We are hoping that the community will try applying our embeddings, or the distances between them, for diverse purposes, perhaps as inputs to some model for a totally different task.
>
> Please look over the new Appendix A, as we believe this provides a solid experimental validation of our method. We believe Appendix A should sufficiently address your main concerns, and hope you will reconsider your score. Please let us know if you have any other questions or comments. Thank you.
>
> \- The authors.

---

### Official Review · AnonReviewer5 · 2020-11-10
**Interesting idea, experimental validation falls short**

**Rating:** 6
**Confidence:** 3

**Review:**

### Overview:

This paper proposes a new acoustic word embedding approach, where acoustic and text embeddings are jointly learned. The text encoder takes either phonemes or characters as input. The novelty of the paper lies in a new loss, which is based on stochastic neighbour embeddings (SNE). The acoustic embedding network is first trained with this loss, after which the text network is trained to produce similar embeddings for matched (acoustic, text) input. The proposed model is evaluated in a word recognition task, where an isolated spoken word's acoustic embedding is compared to the text embeddings and the nearest neighbour is used to classify the spoken word.

**Note the edit at the bottom of this review, based on the authors' feedback.**

### Strengths:

- I do not believe that existing acoustic embedding methods have considered the idea of including a SNE-like objective, especially using the text-view to identify "neigboring" items.
- The use of posteriorgrams as input is well-motivated, in the context of what the paper tries to accomplish.
- The paper is generally easy to follow.


### Weaknesses:

- The models aren't sufficiently compared to previous models on established tasks. (I make this more concrete below.)
- The experiments don't show the effect of using e.g. the posteriorgrams over standard features (like MFFCs). Posteriorgrams means that this approach is essentially reliant on a first-pass ASR model.
- As a minor weakness, some recent related work aren't cited (references given at the bottom).


### Detailed questions and suggestions:

A number of studies in the acoustic word embedding literature have used the same-different task to evaluate performance, e.g. in (He et al., 2017) and [1] and [2]. Given the (somewhat surprising) poor performance of the triplet-based model, I would suggest that the paper does a comparison on the same data and task, to confirm the validity of their triplet-based model. This will also make the work more valuable, in that it can be directly compared to previous studies. One potential issue with the triplet model in this paper is that I believe the model of (He et al., 2017) makes use of cosine similarity, instead of the Euclidean distance. Since labelled examples are available, it would also have been good to compare to a direct classification model, as in [3].

I did not state this as a weakness, since I am worried that I am missing some details, but I am concerned by the overall analysis of Section 4. First, by defining the neighbourhood as in equations (7) and (8), it seems that this model essentially optimises the loss based on whether two acoustic realisations are from the same or different words, and no finer-grained neighbourhood information is included. The only "strength" that is imposed comes from $c_i$, which is essentially linked to the word frequency. The section concludes that "ANE has the added subtlety of pushing or pulling with more 'measured strength' based on how good the embeddings currently are," but if the loss is purely based on a weighing according to the word frequency, could something similar be accomplished by having a type-specific margin for the triplet loss in equation (13)? If I read the analysis correctly the margin is actually completely ignored: "we can ignore the max operator and set $\alpha = 0$ in (13), since they are merely for data selection."  I do not believe that this last statement is correct.

One further suggestion is to look at more advanced sampling strategies in the triplet model, as e.g. in [1] or [6].


### Overall assessment:

Given the shortcomings in the experimental investigation, I do not believe the paper can be accepted as it is. I would recommend that the authors include the above-mentioned additional experiments; this would be a non-trivial extension, and I, therefore, recommend the paper then be submitted a future conference. I award an "Ok but not good enough - rejection".


### Typos, grammar and style:

- "approximate phonetic match task" -> "approximate phonetic matching task"
- "As we can see in Table. 1" -> "As we can see in Table 1"
- "this beautiful equation". I would suggest that the authors remove subjective words such as "beautiful".  (I agree it's a beautiful equation, but I don't think this type of language is appropriate for such a paper.)
- "it is not a good idea". Similar to the above.
- Make sure to write phoneme sequences correctly. See [4].


### Missing references:

1. https://arxiv.org/abs/2006.02295
2. https://arxiv.org/pdf/2006.14007
3. http://arxiv.org/pdf/1510.01032
4. https://arxiv.org/abs/1907.11640
5. https://arxiv.org/abs/2007.00183
6. https://arxiv.org/abs/1804.11297
7. https://arxiv.org/abs/2007.13542


### Edit based on the authors' response

I believe the author(s) were able to address many of the major concerns that I and the other reviewers had.  One issue is that much of this is placed in an appendix, so it doesn't form a core part of the main thread of the paper;  I also disagree about one small point (see my separate comment to the Part 2 message below), but this is minor.  Based on their more extensive investigation, I am changing my rating from "4: Ok but not good enough - rejection" to "6: Marginally above acceptance threshold".

---

> ### Author Response · Authors · 2020-11-19
> **Added Appendix A**
>
> [Part 1/2]
>
> Dear AnonReviewer5, thank you for your helpful review.
>
> We too have been curious about how our method would work with MFCCs. We also agree it is useful to try the word discrimination tasks.
>
> It also seems that the low accuracy of our triplet loss may be at least in part due to the 1-exp(-l2) distance, which we used in a somewhat contrived fashion to get something compatible with ANE's l2 distance. Previous studies consistently report that the cosine works the best with the triplet.
>
> AnonReviewer6 has also suggested some variations in the triplet-based encoders that address some possible inconsistencies in how we compared ANE with the triplet loss.
>
> To this end, we trained a new set of encoders that use MFCCs and the cosine distance (in the case of triplet loss), and ran acoustic word discrimination and cross-view word discrimination experiments similar to those in the previous work [Kamper 2016][Settle 2016][He 2017].
>
> Please see Appendix A where we describe in detail the different triplet-based encoders we trained and the experiments we ran, as we believe this gives our work sufficient grounding with the existing literature. We believe we have reasonable implementations of MFCC-based triplet embeddings in Appendix A, compared in a controlled, even-handed way.
>
> Given these additional results in Appendix A, it doesn't seem necessary that we try again with the same corpus (Switchboard) as the previous studies because our objective in this regard is to highlight some of the differences between the essence of the two basic loss functions (basic triplet & ANE's loss), not rigorously compare with specific embodiments or implementations (we agree it would be nice to do it, but it is not our main objective). It seems our results are sufficient for achieving this goal.

---

> > ### Comment · AnonReviewer5 · 2020-11-19
> > **Addressed many concerns**
> >
> > Thank you for addressing many of my major concerns.  Please see the edit I made at the bottom of my main review, as well as my small comment on Part 2 of your reply (above).  In short, I have changed my assessment from "4: Ok but not good enough - rejection" to "6: Marginally above acceptance threshold".

---

> ### Author Response · Authors · 2020-11-19
> **Part 2 (continued from Part 1)**
>
> [Part 2/2; Continued from Part 1]
>
> \>\> The only "strength" that is imposed comes from c_i, which is essentially linked to the word frequency.
>
> Ignoring everything else, in (12) the first term's coefficient is (essentially) $1-q_{ij}$, whereas in (18) it is just $q_{ij}$. From this alone, you can see that in (12), a high $q_{ij}$ will incur a weak gradient for similar things, where in (18), a high $q_{ij}$ will incur a strong gradient for similar things. But the weak gradient makes more sense, since if $q_{ij}$ is already high for similar things, there is no further reason to change anything -- they're already close, so you shouldn't care anymore; it's better to spend your "dimension budget" (so to speak) on something else.
>
> \>\> could something similar be accomplished by having a type-specific margin for the triplet loss in equation (13)?
>
> Good question. We think this is "partially" possible, in a crude way.
>
> As you increase alpha, you ignore more of the "light offenders" in your data and focus more on the "heavy offenders" in your data (data samples where d(f,g-) is very low and/or d(f,g+) is very high). The "light offenders" are simply dropped from your training data, since the loss for them is 0 and therefore they do not affect your gradient.
>
> So adjusting the alpha could be an indirect means of adjusting your push/pull strength, but in a sort of "binary way" (you don't push/pull at all for light offenders), and it is still more crude than ANE (for the heavy offenders, you're still not weighting in the way ANE does).
>
> Here is the edit we made to Section 4 to discuss this, as well as the data sampling strategies you mentioned:
>
> "However, ANE has the added subtlety of pushing or pulling with more “measured strength” based on how good the embeddings currently are. It is possible that part of this effect could be achieved with the triplet loss as well by adjusting the margin α in (13). We can raise α to focus only on the “heavy offender” triplets (where Sim(f0,gm) is very low and/or Sim(f0,gn) is very high) while ignoring the “light offenders.” Even then, note that the counterintuitive weighting discussed above still holds for the heavy offenders, so the method is still rather crude. Previous studies like (Kamper et al., 2016) searched over a range of values to find the best margin. In this regard, it is possible that ANE is more elegant than the triplet loss in that it can find optimal ways to push and pull with less need for additional manual intervention. Data sampling strategies (Riad et al., 2018) have also been proposed to further refine the accuracy of triplet-based embeddings. Whether such strategies would also benefit ANE is a question we leave to future investigation."
>
> \>\> Typos, grammar and style:
>
> Thank you for these suggestions. We have corrected appropriately.
>
> In regard to the references you listed:
>
> \>\> 1. https://arxiv.org/abs/2006.02295
> \>\> 2. https://arxiv.org/pdf/2006.14007
> \>\> 5. https://arxiv.org/abs/2007.00183
> \>\> 7. https://arxiv.org/abs/2007.13542
>
> The above papers seem to be very recent (less than 6 months old), available in preprint form only. Unless they are of critical relevance to us (they do not seem to be, and core related works by some of the main authors have already been cited), we would prefer to cite them after they have either been peer-reviewed for a conference/journal or reasonably cited by other works.
>
> \>\> 3. http://arxiv.org/pdf/1510.01032
>
> Definitely relevant, especially for Appendix A
>
> \>\> 4. https://arxiv.org/abs/1907.11640
>
> Thank you. This is very interesting. After further web search, we ended up using the bracket notation w/ arpabet per
> http://www.cs.columbia.edu/~julia/courses/CS6998-2019/%5B07%5D%20Phonetics.pdf
>
> \>\> 6. https://arxiv.org/abs/1804.11297
>
> Cited accordingly.
>
> In summary, we believe we have adequately addressed the concerns you expressed. We ask that you examine our Appendix A and other modifications and reconsider your score. Please let us know if you have any follow-up comments or questions. Thank you.
>
> \- The authors

---

> > ### Comment · AnonReviewer5 · 2020-11-19
> > **Addressed detailed comments, small comment**
> >
> > Thank you for addressing these detailed comments.  With regard to the references listed: you are right that some of these are very recent.  But I do think it is useful (and important) to situate the work within the current body of knowledge, especially in regards to recent work.  I also believe that all of the papers listed (apart from https://arxiv.org/abs/2007.00183) have recently been presented in some form at ICASSP or Interspeech 2020, so they have been published in proceedings.

---

> > > ### Author Response · Authors · 2020-11-19
> > > **Added citations**
> > >
> > > Ah, yes, you're right. We did not search thoroughly. Those references appeared in Interspeech & ICASSP, and 2007.00183 also looks interesting & relevant. We have added all of the citations to the paper and re-uploaded.

---

### Official Review · AnonReviewer1 · 2020-11-10
**Interesting idea with promising results**

**Rating:** 6
**Confidence:** 4

**Review:**

This paper proposes a new approach to learn acoustic word embedding by adapting stochastic neighbor embedding (SNE) to sequential inputs. It learns both the acoustic embeddings and text embeddings from two neural encoders. The acoustic word embeddings of two acoustic sequences are learned to be close in Euclidean distance if their transcriptions are the same, or far apart otherwise. The text word embeddings, based on either phoneme or grapheme sequences, are learned to be close to their acoustic word embeddings. The experiment results on an isolated word (name) recognition task show that using nearest-neighbor search alone based on the proposed acoustic and text embeddings in tandem can achieve the same performance as a standard ASR model.

Strong points:
1. Well written with clear presentation.
2. Well motivated by the extension from SNE. And it is technically sound.
3. Interesting idea: Acoustic word embeddings are typically learned with a triplet loss in the literature. The proposed acoustic neighbor embeddings in this paper is a different way but shares similar motivation with the triplet loss, while showing more effective gradients.
4. Promising experimental results: It shows using just the L2 distance between the embedding vector can perform as well as a standard ASR model (without a language model) over large vocabularies in an isolated word recognition task. The paper claims it is the first work to achieve this.

Weak points:
1. The isolated word recognition task requires known word boundaries.  It would be even more useful to show the effectiveness of the proposed approach in continuous speech recognition applications, as in some previous work on acoustic word embeddings, e.g. in lattice rescoring in (Bengio & Heigold, 2014), or in acoustic-to-word ASR in (Settle et al., 2019).
2. Even though it might not affect the isolated word recognition task much, it would be good to also use a language model to build the stronger baseline FST-based ASR model. It would also be interesting to explore how well to combine the proposed embeddings with an LM in an ASR model, especially for continuous speech recognition.
3. Why does the triplet loss perform much worse in Table 2 than in Table 4? More explanation and analysis would be very helpful.
4. Not required, but would be great to have more analyses in the experiments, e.g.:
  - How important is it to use posteriorgrams vs. acoustic features (maybe with a deeper embedding encoder) as the inputs for training the acoustic word embeddings?
  - Analyze individual examples to show the better and worse cases compared to the baselines.
  - Visualization of the proposed embeddings, and possibly with comparison to that of the triplet-loss-trained embeddings.

In general, the paper looks good and I would recommend it for acceptance.

---

> ### Author Response · Authors · 2020-11-19
> **Changes made. Appendix A added.**
>
> Dear AnonReviewer1, thank you for your helpful suggestions.
>
> Application to continuous speech recognition is definitely a direction we wish to explore in future work. However, we are not even sure if speech recognition per se is the best application for ANE. We are hoping that the community will find other interesting uses, either with the embeddings themselves or the distances between embeddings. Our goal in this paper is to introduce ANE and highlight some interesting aspects of it.
>
> In regard to an LM, for most intended applications (named entities like people names, addresses, song titles), the LM is not expected to be more complex than a simple unigram model (with individual prior probabilities for the entities, e.g. based on song popularity, frequency of calling a contact, etc.). Although this is beyond the scope of the paper, incorporating prior probabilities can actually be elegantly done with ANE and the L2 distance. We intend to explore this in future work.
>
> \>\> Why does the triplet loss perform much worse in Table 2 than in Table 4? More explanation and analysis would be very helpful.
>
> We have added this explanation to the paper:
>
> "The phonetic matching task is “easier” than the word recognition task in the sense that we already obtain a fairly accurate phonetic transcription of the speech from the ASR (as opposed to having to recognize the speech from “scratch”), which is probably why the accuracy of the triplet-based embeddings improve in Table 4 over Table 2. On the other hand, there also exists an “accuracy ceiling” because of some out-of-vocabulary words being transcribed far too wrong by the ASR, which is probably why the high accuracy of ANE in Table 2 drops in Table 4."
>
> As other reviewers mentioned, the most pressing need for our paper seems to be further experimental validation, so we have focused our time and effort on adding Appendix A with a new, extensive set of experiments. Please take a look and consider raising your score if you believe significant value has been added. Please let us us know if you have other comments or suggestions. Thank you.
>
> \- The authors

---

### Official Review · AnonReviewer6 · 2020-11-10
**I find the application of SNE to acoustic segments interesting and well-motivated, but I think the presentation would benefit from more direct comparisons as I believe there is misalignment between ANE and multi-view training (vs acoustic-only triplet loss), which could be addressed in additional experiments. I also have concerns that second stage L2-distance training introduces unfair advantages to ANE over multi-view training for comparison on isolated word recognition.**

**Rating:** 6
**Confidence:** 4

**Review:**

Summary:

This work adapts stochastic neighbor embedding (SNE) to acoustic segments to learn “acoustic neighbor embeddings” (ANE), which are fixed dimensional embeddings of variable-length speech. They also learn embeddings of phone or grapheme sequences of words corresponding to these segments. They compare the form of this loss function as well as its gradient with popular multi-view triplet loss approaches. Performance is then given on an isolated word recognition task, shown to outperform triplet losses as well as an FST baseline. Overall, I think there is a slight misalignment between ANE and multi-view training (vs acoustic-only triplet loss), which could be addressed in additional experiments as well as concerns that second stage $L_2$-distance training introduces unfair advantages to ANE over multi-view training for comparison on isolated word recognition.

Pros:
- Interesting analysis of phonetic confusability between words reflected in the $g$ vectors.
- I appreciate the straightforward nature of the extension, as well as comparison and motivations versus (multi-view) triplet losses in Section 4.
- This approach is strong for the isolated word recognition task versus the provided baselines (FST and triplet).

Comments:
- It seems like the main comparison between loss functions is being done between ANE and a multi-view triplet loss, which is okay, but I think the better comparison would be between triplet losses that only operate on acoustic segments (e.g. Kamper et al https://arxiv.org/pdf/1510.01032.pdf). It looks like you’re training a new acoustic embedding function (similar more to acoustic-only triplet losses than to multi-view triplet losses) and then incorporating the second view $g$ through a secondary $L_2$ loss applied after first training ANE. You could compare the pipelines of acoustic-only triplet loss versus ANE to learn f followed by either an $L_2$-distance or multi-view training approach to learn g (with or without f fixed). Also, since you’re using nearest neighbors to perform your evaluation, fine-tuning after training the multi-view approach with a second stage $L_2$-distance training would be fairer given that that’s what’s being used through nearest neighbor search at evaluation time I believe. It may detract from the overall aesthetic of moving away from triplet loss, but I think these would be reasonable comparisons. ANE shows a well-motivated method for learning an acoustic embedding space, but incorporation of the second view through $L_2$-distance versus a triplet loss isn’t compared. Additionally, an argument can be made for establishing $g_j$  an $f_j$ as roughly the same from Equation 14, but in practice it appears that the differences in these values actually help stabilize training as g’s are consistent for a particular word rather than varying for every acoustic instance. I admit this hasn’t been adequately/explicitly shown in prior work, but I wanted to mention it as another benefit to comparison directly with acoustic-only triplet losses which would remove this approximation in your derivations.

- It would be nice to see additional results on some common acoustic word embedding datasets, or evaluations that include the word discrimination task using average precision (AP) in addition to word accuracy. Also, there are many applications of these acoustic embedding approaches to low-resource domains in which word-labels may not be available but clusterings of similar examples may be. It could be interesting to give a word discrimination evaluation (in AP) for just the acoustic embedding outputs (from $f$) for this reason.

- Additional evaluation using typical acoustic features would be nice for curiosity reasons as well as due to the above concern about common applications to low-resource domains where high-performing ASR systems may not exist a priori.

Overall:
I find application of SNE to acoustic segments interesting, but I think the presentation it could benefit from more direct comparisons. In particular the three sets of experiments I would have in mind would be:

- acoustic-only triplet loss
    - train $f$ with acoustic-only triplet loss, then train $g$ with $L_2$-distance as given in Eq 9
    - train $f$ with acoustic-only triplet loss, then train $g$ with multi-view approach as given in Eq 13 (with $f$ fixed)
    - train $f$ with acoustic-only triplet loss, then train $g$ with multi-view approach as given in Eq 13 (without $f$ fixed)
- ANE experiments
    - train $f$ with ANE loss, then train $g$ with $L_2$-distance as given in Eq 9
    - train $f$ with ANE loss, then train $g$ with multi-view approach as given in Eq 13 (with $f$ fixed)
    - train $f$ with ANE loss, then train $g$ with multi-view approach as given in Eq 13 (without $f$ fixed)
- Multi-view experiments
    - train $f$ and $g$ with multi-view approach as given in Eq 13
    - train $f$ and $g$ with multi-view approach as given in Eq 13, then fine-tune $g$ with $L_2$-distance as given in Eq 9

In addition, an AP evaluation (and, to a lesser extent given that time is limited, more common acoustic embedding datasets or domains) could be nice to include.

---

> ### Author Response · Authors · 2020-11-19
> **Added Appendix A with more experiments**
>
> Dear AnonReviewer6, thank you for your helpful review.
>
> We had actually been wondering about some of the possibilities you suggested, of adding a "further fine-tuning" step for the g obtained by multi-view training, as well as doing an "acoustic-only" triplet-loss training, so we are happy you raised these issues, and we are glad to follow up.
>
> You have suggested 8 different experiments. Out of these, it seems that the 1st and 8th should be enough to address your concerns, i.e., the misalignment between ANE & multi-view, and the possible unfair advantage for ANE in word recognition. Is this correct?
>
> It seems that if we did the following two experiments:
>
> (Your suggestion #1) train f with acoustic-only triplet loss per [Kamper 2016], then train g with L2-distance as given in Eq 9 (-> addresses the "misalignment" issue)
>
> (Your suggestion #8) train f and g with multi-view approach as given in Eq 13, then fine-tune g with L2-distance as given in Eq 9 (-> addresses the "unfair advantage" issue)
>
> Above, #1 would be a more direct parallel to the ANE-based method, since it follows the same general notion of training f independently, then training g while keeping f fixed, hence addressing the "misalignment" issue. And #8 would add an additional step of fine-tuning g with a training criterion that explicitly tries to bring it close to f, hence addressing the "unfair advantage" issue. Do you agree?
>
> We also agree that it is useful to try the word discrimination tasks that were conducted by the previous studies.
>
> Furthermore, the other reviewers have suggested trying with MFCC instead of posteriorgram inputs, as well as using the cosine distance with the triplet loss, since previous studies (such as [Kamper 2016]) reported that the cosine distance gave the best accuracy.
>
> Therefore, to address all these concerns while avoiding a combinatorial explosion of experiments, we prepared a new set of data and trained the encoders per your suggestions #1 & #8 but using _MFCCs_ and the _cosine_ distance (instead of L2).
>
> Specifically, per your suggestion #1 we trained "mfc-tripf-f", followed by "mfc-tripf-cos-g". Per your suggestion #8, we trained "mfc-tripfg-f"+"mfc-tripfg-g", followed by an additional fine-tuning of "mfc-tripfg-g" to produce "mfc-tripfg-cos-g".
>
> To compare with ANE, we also trained from scratch an ANE f encoder and g encoder that uses MFCCs, "mfc-ane-f"+"mfc-ane-g". We must still use the L2 distance with ANE, since the fundamental design of SNE uses the L2 distance (it may be possible to train a cosine-distance ANE, but that is definitely beyond the scope of this paper).
>
> We also conducted acoustic word discrimination tasks (as in [Kamper 2016, Settle 2016, He 2017]) as well as the cross-view word discrimination task [He 2017] using these embeddings, on a set of test data with a similar size as the one used in the previous work (60M pairs, 97K "same pronunciation" pairs).
>
> Due to the large number of experiments, it was impossible to describe this additional work in one additional page as allowed by ICLR, so we added an Appendix to the paper with a full description.
>
> \>\> Additionally, an argument can be made for establishing g_j and f_j as roughly the same from Equation 14, but in practice it appears that the differences in these values actually help stabilize training as g’s are consistent for a particular word rather than varying for every acoustic instance. I admit this hasn’t been adequately/explicitly shown in prior work
>
> So in regard to this, as we mentioned in our "General notes and summary of revisions" post above, we found our single-view [Kamper 2016] mfc-tripf-f to give better accuracy than the cross-view [He 2017] mfc-tripfg-f, and we believe this result is OK, as we explained above.
>
> We hope the new Appendix A addresses your concerns sufficiently and that you would consider raising your score. Please let us know if you have any further comments or questions. Thank you.
>
> \- The authors

---

### Author Response · Authors · 2020-11-19
**General notes and summary of revisions**

We wish to thank all 5 reviewers for their very helpful reviews.

We have responded to each reviewer individually, and these are some general notes and summary of changes:

1. We have added a new Appendix A, where we trained a new set of encoders (reflecting some excellent suggestions from AnonReviewer6) that use MFCC inputs, applied the cosine distance to the triplet loss, added extensions to the training criteria to enhance compatibility with ANE, and also conducted acoustic word discrimination and cross-view word discrimination tasks similar to those done in previous work. We are confident that we have made reasonable implementations of the triplet loss, made even-handed comparison between different methods, and that our experimental validation is sound.

2. We tweaked some of the wording to avoid the misperception that we are trying to directly compete with specific embodiments or implementations of the triplet loss (it would be nice to do so, but it is not an objective of this paper). One of the main goals of our paper is to highlight some of the fundamental differences between the "essence" of ANE's loss function and the triplet loss function. Given that they both seem to share the same idea of pulling together similar things and pushing apart different things, how do they differ? This is the question we are more interested in.

3. We made some clarifications that the state-of-the-art in multi-view triplet-based embedding actually uses two loss functions (obj0+obj2 in [He 2017]), but we consider this as out of scope for our paper (see discussion in Appendix A)

4. One point we wish to raise to all reviewers: We found in our experiments that the multi-view triplet [He 2017] can actually underperform the single-view triplet [Kamper 2016, Settle 2016] when only one loss function (obj0) is used. The state-of-the-art in [He 2017] is when using _two_ loss functions (obj0+obj2), but when obj0 is used alone, it is unclear from [He 2017] whether obj0 compares better or worse than the single-view of [Settle 2016]. (It is actually also unclear to us, by the way, how fairly the AP of obj0+obj2 in Table 2 is being compared to the single-view[Settle 2016] AP; how similar were the training processes & conditions for the two different loss types that produced AP values of 0.671 and 0.806?)

    We think it makes sense that obj0 alone can sometimes underperform, because the single-view triplet loss and multi-view triplet loss are essentially the same loss function. The difference is that the single-view training is "easier" because it only needs to train f, whereas the multi-view training has the added "burden" of training both f and g simultaneously. Wouldn't there be cases where the single-view finds a better minimum more easily? Based on this notion, we are comfortable with our result.

5. We added comments that the 1-exp(-L2) triplet distance we used is somewhat contrived (it was used because we wanted something compatible with ANE's L2 distance), and previous studies actually reported the cosine distance as working best, which could be the reason why the accuracy is low for our triplet implementation. We have pointed readers to Appendix A for further experiments using the cosine distance for the triplet distance.

6. Since our first submission, we found that our triplet-loss-based embeddings improve if we separately generate additional training triplets than directly obtain them from the same microbatches used by ANE training, and updated the values in Tables 2 and 4 accordingly (they are still much lower than ANE, so the overall picture does not change).

7. Other edits have been made following reviewer comments, such as more description of the phonetic matching results, details on ASR, clarification of the different FST decoders used, comments on adjusting the triplet loss margin, phone sequence notation, etc.

---

### Decision · Program_Chairs · 2021-01-07
**Final Decision**

**Decision:**

Reject

**Comment:**

This paper propose to learn the embedding of audio segment in the framework of stochastic neighbor embedding (SNE), where the embeddings of the same word shall be close to each other. The method was initially demonstrated for name recognition. The use of SNE for acoustic embedding is novel and this is recognized by all reviewers. There has been quite some discussions between the authors and reviewers/AC, and the papers has got improved since. To summarize:
1. The discussion of the properties of SNE (the reduction from SNE to weighted least squares terms) was not accurate and confused multiple reviewers. The authors have made some clarifications and added citations. As the author claim this to be a contribution, I feel this part of the main text can be further strengthened and made more accurate.
2. For the experiment in main paper, the comparison between proposed method and prior work was not fair since the proposed method use outputs from an ASR system to obtain phone posteriors. The authors then added more results for the word discrimination task in the appendix. But reviewers are still concerned that the authors are not comparing with the strongest variant of He et al, 2017, and that comparisons are shown for embeddings with low dimensions. The reviewers believe this set of experiments shall be more illuminating, and be moved to the main text.
3. The biggest concern at the intuition level is whether it is the best choice to make the affinity binary, which does not take into account the more fine-grained similarity between different words. Quoting the comment from Reviewer 6:  "The argument of trying to be as simple as possible is reasonable, but we would have liked to see it motivated by some experiments. Something along the lines of a method which introduced rudimentary edit distance-based affinities and then presented their version with hard affinities, and then show some results comparing them. These edit distance-based versions could be as simple or complicated as they felt necessary, but it would be nice for some comparison to be made in order to then dismiss them."

Overall, we think the paper is borderline in the current stage, and the paper can be further improved if the above concerns are properly addressed.